# COMPOSITIONAL OBVERTER COMMUNICATION LEARNING FROM RAW VISUAL INPUT

**Edward Choi** *
Georgia Institute of Technology
Atlanta, GA, USA
mp2893@gatech.edu

**Angeliki Lazaridou & Nando de Freitas**
DeepMind
London, UK
{angeliki, nandodefreitas}@google.com

## ABSTRACT

One of the distinguishing aspects of human language is its compositionality, which allows us to describe complex environments with limited vocabulary. Previously, it has been shown that neural network agents can learn to communicate in a highly structured, possibly compositional language based on disentangled input (*e.g.* hand-engineered features). Humans, however, do not learn to communicate based on well-summarized features. In this work, we train neural agents to simultaneously develop visual perception from raw image pixels, and learn to communicate with a sequence of discrete symbols. The agents play an image description game where the image contains factors such as colors and shapes. We train the agents using the *obverter* technique where an agent introspects to generate messages that maximize its own understanding. Through qualitative analysis, visualization and a zero-shot test, we show that the agents can develop, out of raw image pixels, a language with compositional properties, given a proper pressure from the environment.

## 1 INTRODUCTION

One of the key requirements for artificial general intelligence (AGI) to thrive in the real world is its ability to communicate with humans in natural language. Natural language processing (NLP) has been an active field of research for a long time, and the introduction of deep learning (LeCun et al., 2015) enabled great progress in NLP tasks such as translation, image captioning, text generation and visual question answering (Cho et al., 2014; Bahdanau et al., 2014; Vinyals et al., 2015; Karpathy & Fei-Fei, 2015; Hu et al., 2017; Serban et al., 2016; Lewis et al., 2017; Antol et al., 2015). However, training machines in a supervised manner with a large dataset has its limits when it comes to communication. Supervised methods are effective for capturing statistical associations between discrete symbols (*i.e.* words, letters). The essence of communication is more than just predicting the most likely word to come next; it is a means to coordinate with others and potentially achieve a common goal (Austin, 1975; Clark, 1996; Wittgenstein, 1953).

An alternative path to teaching machines the art of communication is to give them a specific task and encourage them to learn how to communicate on their own. This approach will encourage the agents to use languages grounded to task-related entities as well as communicate with other agents, which is one of the ways humans learn to communicate (Bruner, 1981). Recently, there have been several notable works that demonstrated the emergence of communication between neural network agents. Even though each work produced very interesting results of its own, in all cases, communication was either achieved with a single discrete symbol (as opposed to a sequence of discrete symbols) (Foerster et al., 2016; Lazaridou et al., 2017) or via a continuous value (Sukhbaatar et al., 2016; Jorge et al., 2016). Not only is human communication un-differentiable, but also using a single discrete symbol is quite far from natural language communication. One of the key features of human language is its compositional nature; the meaning of a complex expression is determined by its structure and the meanings of its constituents (Frege, 1892). More recently, Mordatch & Abbeel (2017) and Kottur et al. (2017) trained the agents to communicate in grounded, compositional language. In both studies, however, inputs given to the agents were hand-engineered features (*disentangled input*) rather than raw perceptual signals that we receive as humans.

---

*Work done as an intern at DeepMind.

In this work, we train neural agents to simultaneously develop visual perception from raw image pixels, and learn to communicate with a sequence of discrete symbols. Unlike previous works, our setup poses greater challenges to the agents since visual understanding and discrete communication have to be induced from scratch in parallel. We place the agents in a two-person image description game, where images contain objects of various color and shape. Inspired by the pioneering work of Batali (1998), we employ a communication philosophy named *obverter* to train the agents. Having its root in the theory of mind (Premack & Woodruff, 1978) and human language development (Milligan et al., 2007), the obverter technique motivates an agent to search over messages and generate the ones that maximize their own understanding. The contribution of our work can be summarized as follows:

- We train artificial agents to learn to disentangle raw image pixels and communicate in compositional language at the same time.
- We describe how the obverter technique, a differentiable learning algorithm for discrete communication, could be employed in a communication game with raw visual input.
- We visualize how the agents are perceiving the images and show that they learn to disentangle color and shape without any explicit supervision other than the communication one.
- Experiment results suggest that the agents could develop, out of raw image input, a language with compositional properties, given a proper pressure from the environment (*i.e.* the image description game).

Finally, while our exposition follows a multi-agent perspective, it is also possible to interpret our results in the single-agent setting. We are effectively learning a neural network that is able to learn disentangled compositional representations of visual scenes, without any supervision. Subject to the constraints imposed by their environment, our agents learn disentangled concepts, and how to compose these to form new concepts. This is an important milestone in the path to AGI.

## 2 METHOD

### 2.1 THE TWO-PERSON IMAGE DESCRIPTION GAME

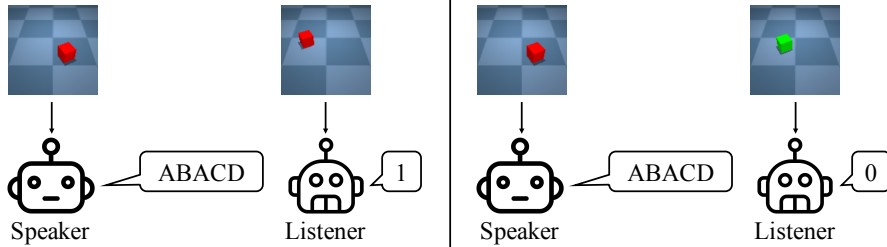

Figure 1: The two-person image description game. Speaker observes an image and generates a message (*i.e.* a sequence of discrete symbols). The listener, after observing a separate image and the message, must correctly decide whether it is seeing the same object as the speaker (left side; output 1) or not (right side; output 0).

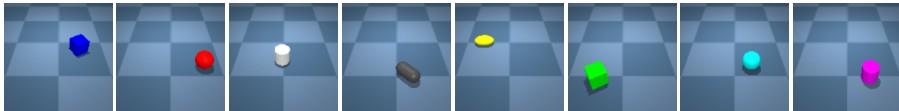

Figure 2: Example images of the dataset. There are eight colors (*blue, red, white, gray, yellow, green, cyan, magenta*), and five shapes (*box, sphere, cylinder, capsule, ellipsoid*), giving us total 40 combinations.

We choose a straightforward image description game with two factors (color and shape) so that we can perform extensive analysis on the outcome confidently, based on full control of the experiment. In a single round of the two-person image description game, one agent becomes the speaker and the other

the listener. The speaker is given a random image, and generates a message to describe it. The listener is also given a random image, possibly the same image as the speaker's. After hearing the message from the speaker, the listener must decide if it is seeing the same object as the speaker (Figure 1). Note that an *image* is the raw pixels given to the agents, and an *object* is the thing described by the image. Therefore two different images can depict the same object. In each round the agents change roles of being the speaker and the listener.

We generated synthetic images using Mujoco physics simulator[1]. The example images are shown in Figure 2. Each image depicts a single object with a specific color and shape in 128×128 resolution. There are eight colors (*blue, red, white, gray, yellow, green, cyan, magenta*) and five shapes (*box, sphere, cylinder, capsule, ellipsoid*), giving us 40 combinations. We generated 100 variations for each of the 40 object type. Note that the position of the object varies in each image, changing the object size and the orientation. Therefore even if the speaker and the listener are given the same object type, the actual images are very likely to be different, preventing the agents from using pixel-specific information, rather than object-related information to win the game.

## 2.2 MODEL ARCHITECTURE

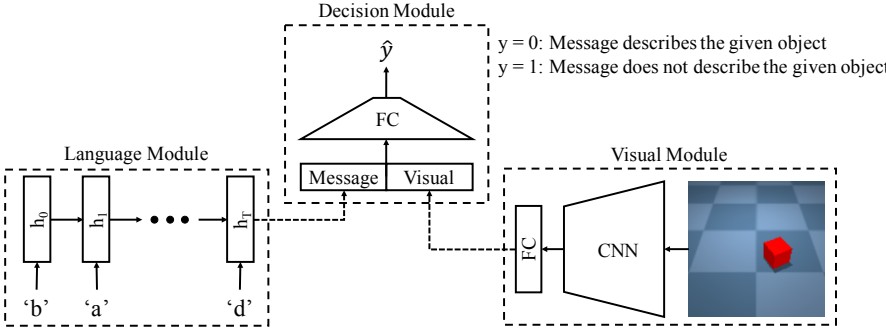

Figure 3: Agent model architecture. The visual module processes the image, and the language module generates or consumes messages. The decision module accepts embeddings from both modules and produces the output. The solid arrows indicate modifying the output from the previous layer. The dotted arrows indicate copying the output from the previous layer.

Aside from using disentangled input, another strong assumption made in previous works(Batali, 1998; Mordatch & Abbeel, 2017) was that the agents had access to the true intention of the speaker. In Batali (1998), the listener was trained to modify its RNN hidden vector as closely to the speaker's intention (meaning vector; please see Table 5 in Appendix A) as possible. In Mordatch & Abbeel (2017), each agent had an auxiliary task to predict the goals of all other agents. In both cases, the true meaning/goal vector was used to update the model parameters, exposing the disentangled information to the agents. In order to relax this assumption and encourage the agents to develop communication with minimal guidance, our model uses no other signal than whether the listener made a correct decision.

Figure 3 depicts the agent model architecture. We use a convolutional neural network followed by a fully-connected layer to process the image. A single RNN, specifically the gated recurrent units (GRU) (Cho et al., 2014), is used for both generating and consuming messages (the message generation using the obverter strategy is described in the next section). When consuming a message, the image embedding from the visual module and the message embedding from the language module are concatenated and processed by another fully-connected layers (*i.e.* decision module) with the sigmoid output $\hat{y}$, 0 being "My (listener) image is different from the speaker's" and 1 being "My image is the same as the speaker's". Further details of the model architecture (*e.g.* number of layers) are described in Appendix C.

---

[1]http://www.mujoco.org/index.html

## 2.3 OBVERTER TECHNIQUE

Although our work is inspired by Batali (1998) (see Appendix A for the description of Batali (1998)), *obverter* technique is a general message generation philosophy used/discussed in a number of communication and language evolution studies (Hurford, 1989; Oliphant & Batali, 1997; Smith, 2001; Kirby & Hurford, 2002), which has its root in the theory of mind. Theory of mind (Premack & Woodruff, 1978) observes that a human has direct access only to one's own mind and not to the others'. Therefore we typically assume that the mind of others is analogous to ours, and such assumption is reflected in the functional use of language (Bruner, 1981). For example, if we want to convey a piece of information to the listener[2], it is best to speak in a way that maximizes the listener's understanding. However, since we cannot directly observe the listener's state of mind, we cannot exactly solve this optimization problem. Therefore we posit that the listener's mind operates in a similar manner as ours, and speak in a way that maximizes *our understanding*, thus approximately solving the optimization problem. This is exactly what the obverter technique tries to achieve.

When an agent becomes the teacher (*i.e.* speaker), the model parameters are fixed. The image is converted to an embedding via the visual module. After initializing its RNN hidden layer to zeros, the teacher at each timestep evaluates $\hat{y}$ for all possible symbols and selects the one that maximizes $\hat{y}$. The RNN hidden vector induced by the chosen symbol is used in the next timestep. This is repeated until $\hat{y}$ becomes bigger than the predefined threshold, or the maximum message length is reached (see Appendix D for algorithm). Therefore the teacher, through introspection, greedily selects characters at each timestep to generate a message such that the consistency between the image and the message is as clear to itself as possible. When an agent becomes the learner (*i.e.* listener), its parameters are updated via back-propagating the cross entropy loss between its output $\hat{y}$ and the true label $y$. Therefore the agents must learn to communicate only from the true label indicating whether the teacher and the learner are seeing the same object.

We remind the reader that only the learner's RNN parameters are updated, and the teacher uses its fixed RNN. Therefore an agent uses only one RNN for both speaking and listening, guaranteeing self-consistency (see Appendix B for a detailed comparison between the obverter technique and the RL-based approach). Furthermore, because the teacher's parameters are fixed, message generation can easily be extended to be more exploratory. Although in this work we deterministically selected a character in each timestep, one can, for example, sample characters proportionally to $\hat{y}$ and still use gradient descent for training the agents. Using a more exploratory message generation strategy could help us discover a more optimal communication language when dealing with complex tasks.

Another feature of the obverter technique is that it observes the *principle of least effort* (Zipf, 1949). Because the teacher stops generating symbols as soon as $\hat{y}$ reaches the threshold, it does not waste any more effort trying to *perfect* the message. The same principle was implemented in one way or another in previous works, such as choosing the shortest among the generated strings (Kirby & Hurford, 2002) or imposing a small cost for generating a message (Mordatch & Abbeel, 2017).

## 2.4 ENVIRONMENTAL PRESSURE FOR COMPOSITIONAL COMMUNICATION

During the early stages of research, we noticed that randomly sampling object pairs (one for the teacher, one for the learner) lead to agents focusing only on colors and ignoring shapes. When the teacher's object is fixed, there are 40 (8 colors × 5 shapes) possibilities on the learner's side. If the teacher only talks about the color of the object, the learner can correctly decide for 36 out of 40 possible object types. The learner makes incorrect decisions only when the teacher and the learner are given objects of the same color but different shapes, resulting in $90\%$ accuracy on average. This is actually what we observed; the accuracy plateaued between $0.9$ and $0.92$ during the training, and the messages were more or less the same for objects with the same color. Therefore when constructing a mini-batch of images, we set 25% to be the object pairs of the same color and shape, 30% the same shape but different colors, 20% the same color but different shapes. The remaining 25% object pairs were picked randomly[3].

Vocabulary size (*i.e.* number of unique symbols) and the maximum message length were also influential to the final outcome. We noticed that a larger vocabulary and a longer message length

---

[2]We assume the listener cannot speak but only listen. Therefore the listener's mind is completely hidden.

[3]Other ratios could equally work well, but we empirically found this ratio to be effective in many experiments

helped the agents achieve a high communication accuracy more easily. But the resulting messages were more challenging to analyze for compositional patterns. In all our experiments we used 5 and 20 respectively for the vocabulary size and the maximum message length, similar to what Batali (1998) used. This suggests that the environment plays as important, if not more, role as the model architecture in the emergence of complex communication as discussed by previous studies (Kirby et al., 2014; Bratman et al., 2010; Kottur et al., 2017) and should be a main consideration for future efforts. Further details regarding hyperparameters are described in Appendix E.

## 3 EXPERIMENTS

In this section, we first study the convergence behavior during the training phase. Then we analyze the language developed by the agents in terms of compositionality. As stated in the introduction, in compositional language, the meaning of a complex expression is determined by its structure and the meanings of its constituents. With this definition in mind, we focus on two aspects of the inter-agent communication to evaluate its compositional properties: the structure (*i.e.* grammar) of the communication, and zero-shot performance (*i.e.* generalizing to novel stimuli). These two aspects, which are both necessary conditions for any language to be considered compositional, have been used by previous works to study the compositional nature of artificial communication (Batali, 1998; Mordatch & Abbeel, 2017; Kottur et al., 2017).

To evaluate the *structure* of the messages, we study the evolution of the communication as training proceeds, and try to derive a grammar for expressing colors and shapes. To evaluate the *zero-shot* capabilities, we test if the agents can compose consistent messages for objects they have not seen during the training. Moreover, we visualize the image embeddings from the visual modules of both agents to understand how they are recognizing colors and shapes, the results of which, for a better view of the figures, are provided in Appendix H.

### 3.1 CONVERGENCE BEHAVIOR

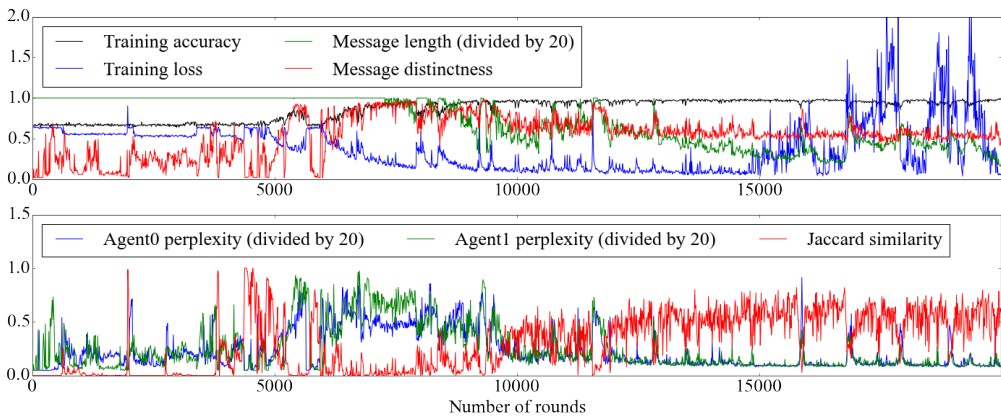

Figure 4: Progress during the training (best seen in color). (Top) We plot the training accuracy, training loss, average message length and average message distinctness in each round. (Bottom) We plot the perplexities and the Jaccard similarity of the messages spoken by both agents in each round. Note that the average message length and the perplexities are divided by 20 to match the y-axis range with other metrics.

Figure 4 shows the convergence behavior during the training. Training accuracy was calculated by rounding the learner's sigmoid output by 0.5. Message distinctness was calculated by dividing the number of unique messages in the mini-batch by the size of the mini-batch. Ideally there should be, on average, 40 distinct messages in the mini-batch of 50 images, therefore giving us 0.8 distinctness. Every 10 round, both agents were given the same 1,000 randomly sampled images to generate 1,000 message pairs. Then perplexity was calculated for each object type and averaged, thus indicating the average number of distinct messages used by the agents to describe a single object type (note that

perplexities in the plot was divided by 20). Jaccard similarity between both agents' messages was also calculated for each object type and averaged.

At the beginning, the listener (*i.e.* learner) always decides it is not seeing the same object as the speaker, giving us $0.75$ accuracy[4]. But after $7,000$ rounds, accuracy starts to go beyond $0.9$. Loss is negatively correlated with accuracy until round $15,000$, where it starts to fluctuate. Accuracy, however, remains high due to how accuracy is measured; by rounding the learner's output by $0.5$. Although we could occasionally observe some patterns in the messages when both accuracy and loss were high, a lower loss generally resulted in a clearer communication structure (*i.e.* grammar) and better zero-shot performance. The loss fluctuation also indicates some instability in the training process, which is a potential direction for future work. Message distinctness starts at near $0$, indicating the agents are generating the same message for all object types. After round $7,000$, where both message distinctness and message length reach their maximum, both start to decrease. But message distinctness never goes as high as the ideal $0.8$, meaning that the agents are occasionally using the same message for different object types, as will be shown in the following section.

Both perplexities and Jaccard similarity show seemingly meaningless fluctuation at early rounds. After round $7,000$, perplexities and Jaccard similarity show negatively correlated behavior, meaning that not only is each agent using consistent messages to describe each object type, but also both agents are using very similar messages to describe each object type. We found perplexity and Jaccard similarity to be an important indicator of the degree of the communication structure. During rounds $7,000 \sim 8,000$, performance was excellent in terms of loss and accuracy, but perplexity was high and Jaccard similarity low, indicating the agents were assigning incoherent strings to each object type just to win the game. Similar behavior was observed in the early stages of language evolution simulation in Kirby & Hurford (2002) where words represented some meanings but had no structure (*i.e.* protolanguage). It seems that artificial communication acquires compositional properties after the emergence of protolanguage regardless of whether the input is entangled or disentangled.

## 3.2 GRAMMAR ANALYSIS

We choose agents from different training rounds to highlight how the language becomes more structured over time. Table 1 shows agents' messages in the beginning (round 40), when the training accuracy starts pushing beyond $90\%$ (round $6,940$), when agents settle on a common language (round $16,760$).

In round $40$, both agents are respectively producing the same message for all object types as mentioned in section 3.1. We might say the messages are structured, but considering that the listener always answers $0$ in early rounds, we cannot say the agents are *communicating*. In round $6,940$, which is roughly when the agents begin to communicate more efficiently, training accuracy is significantly higher than round $40$. However, perplexities show that both agents are assigning many names to a single object type (40-80 names depending on the object type), indicating that the agents are focusing on pixel-level differences between images of the same object type. Table 1 shows, as an example, the messages used by both agents to describe the *red sphere*. Due to high perplexity, it is difficult to capture the underlying grammar of the messages even with regular expression. Furthermore, as Jaccard similarity indicates, both agents are generating completely different messages for the same object type. In round $16,760$, as the perplexities and Jaccard similarity tell us, the agents came to share a very narrow set of names for each object type (1-4 names depending on the object type). Moreover, the names of the same-colored objects and same-shaped objects clearly seem to follow a pattern. Overall, each of the three phases (round 40, round 6,940, round 16,760) seem to represent the development of visual perception, learning to communicate, and emergence of structure.

We found the messages in round $16,760$ could be decomposed in a similar manner as Table 6 in Appendix A. The top of Table 2 shows a possible decomposition of the messages from round $16,760$ and the bottom shows the rules for each color and shape derived from the decomposition. According to our analysis, the agents use the first part of the message (*i.e.* prefix) to specify a shape, and the second part (*i.e.* suffix) to specify a color. However, they use two different strings to specify a shape. For example, the agents use either *aaaa* or *bbbbb* to describe a *box*. The strings used for specifying colors show slightly weaker regularity. For example, *red* is always described by either the suffix c or suffix e, but *magenta* is described by the suffix bb, bd, and sometimes b or bc. ā used for *gray* objects

---

[4]Note that we set $25\%$ of the mini-batch to be objects of the same color and shape.

**Round 40**

(Training accuracy:66.1%, Agent0 perplexity:1.0, Agent1 perplexity:1.0, Jaccard similarity:0.0)

| - | All objects described by Agent 0 | All objects described by Agent 1 |
|---|---|---|
| **Message** | dddddddddddddddddddd | bbbbbbbbbbbbbbbbbbbb |

**Round 6,940**

(Training accuracy:93.1%, Agent0 perplexity:9.90, Agent1 perplexity:17.73, Jaccard similarity:0.0)

| - | Red sphere described by agent 0 | Red sphere described by agent 1 |
|---|---|---|
| **Messages** | aeceaeaeaaaeeeeeeeeee | aaedacacaaaaaaaaaaaa |
|  | aacceeaaaaeeeeeeeeeee | aaccdaacaaaaaaaaaaaa |
|  | aaccceaaaaaaaaaeeeeee | aaabcdadaaaaaaaaaaaa |
|  | aeeeeaaaaeeeeeeeeeeee | aaeeacaeaaaaaaaaaaaa |
|  | aeaceeaeaeeeeeeeeeeee | aaccdadaaaaaaaacaaaa |
|  | aceacacaaaaaaeeeeeeee | aaedaceaaaaaaaaaaaaa |
|  | abeeeeaeeeeeeeeeeeeee | aaeaccaeaaaaaaaaaaaa |
|  | aacceeaeeeeeeeeeeeeee | aaceacaacaaaaaaaaaaa |
|  | aacceeaeaeeeeeeeeeeee | aacdacdaaaaaaaaaaaaa |
|  | aeeacacaaaeeeccceeeee | aaccdadaaaaacaaaaaaa |

**Round 16,760**

(Training accuracy:98.0%, Agent0 perplexity:1.69, Agent1 perplexity:1.62, Jaccard similarity:0.82)

| - | Box | Sphere | Cylinder | Capsule | Ellipsoid |
|---|---|---|---|---|---|
| **Blue** | bbbbbbb{b,d} | bb{b,d} | bbbbbb{b,d} | bbbbb{c,d} | bbbb{_,c,d} |
| **Red** | aaaa{c,e} | aa{c,e} | aaa{c,e} | a{c,e} | c,e |
| **White** | bbbbbb | b,d | bbbb{b,d} | bbb{b,d} | bb{_,c,d} |
| **Gray** | aaa | a | aa | c | b,bd |
| **Yellow** | aaaaaa | aaaa | aaaaa | aaa | a{a,e} |
| **Green** | aaaa{a,ad} | aa{a,ad} | aaa{a,ad} | a{a,ad} | a |
| **Cyan** | a×20 | bbb{b,d} | bbbbbb{b,d} | bbbbbbd | bbbbb{b,d} |
| **Magenta** | bbbbbb | b{b,d} | bbbb{b,d} | bbbbd | bbb{_,c,d} |

Table 1: (Top) Messages used by both agents when speaking about any object in round 40. (Middle) Ten most frequent messages used by each agent to describe a *red sphere* in round 6, 940. (Bottom) Messages most often used by both agents for each object type in round 16, 760. Brackets indicate the variation often seen at the last character. Underscores indicate blanks.

represents deletion of the prefix *a*. Note that removing prefixes beyond their length causes the pattern to break. For example, *gray box, gray sphere* and *gray cylinder* use the same āāa to express the color, but *gray capsule* and *gray ellipsoid* use irregular suffixes.

Despite some irregularities and one exceptional case (*cyan box*), the messages provide strong evidence that the agents learned to properly recognize color and shape from raw pixel input (see Appendix H for studying what the visual module learned), mapped each color and shape to prefixes and suffixes, and are able to compose meaningful messages to describe a given image to one another. Communication accuracy for each object type is described in Appendix F. Communication examples and their analysis are given in Appendix G.

## 3.3 ZERO-SHOT EVALUATION

If the agents have truly learned to compose a message that can be divided into a color part and a shape part, then they should be able to accurately describe an object they have not seen before, which is another necessary condition for a compositional language. Therefore, we hold out five objects (the shaded cells in Table 3) from the dataset during the training and observe how agents describe five novel objects during the test phase. The agents were chosen from round 19, 980, which showed a high accuracy (97.8%), low perplexities (1.48, 1.65) and a high Jaccard similarity (0.75). Table 3 shows a potential decomposition of the messages used by the agents (original messages are described by Table 8 in Appendix I). We can observe that there is clearly a structure in the communication, although some messages show somewhat weaker patterns compared to when the agents were trained with all object types (Table 2). Suffixes for specifying *yellow* and *magenta* are especially irregular,

| - | Box | Sphere | Cylinder | Capsule | Ellipsoid |
|---|---|---|---|---|---|
| **Blue** | *bbbbb* bb{b,d} | bb{b,d} | *bbbb* bb{b,d} | *bbb* bb{c,d} | *bb* bb{_,c,d} |
| **Red** | *aaaa* {c,e} | *aa* {c,e} | *aaa* {c,e} | *a* {c,e} | {c,e} |
| **White** | *bbbbb* b | {b,d} | *bbbb* {b,d} | *bbb* {b,d} | *bb* {_,c,d} |
| **Gray** | *aaaa* āāa | *aa* āāa | *aaa* āāa | *a* āāc | āā{b,bd} |
| **Yellow** | *aaaa* aa | *aa* aa | *aaa* aa | *a* aa | a{a,e} |
| **Green** | *aaaa* {a,ad} | *aa* {a,ad} | *aaa* {a,ad} | *a* {a,ad} | a |
| **Cyan** | a×20 | bbb{b,d} | *bbbb* bbb{b,d} | *bbb* bbbd | *bb* bbb{b,d} |
| **Magenta** | *bbbbb* bb | b{b,d} | *bbbb* b{b,d} | *bbb* bd | *bb* b{_,c,d} |

Define *Blue, White, Cyan, Magenta* as color group 0, rest as color group 1.

| - | Rule |
|---|---|
| **Blue** | End with bbb or bbd |
| **Red** | End with c or e |
| **White** | End with b or d |
| **Gray** | End with āāa |
| **Yellow** | End with aa |
| **Green** | End with a or ad |
| **Cyan** | End with bbbb or bbbd |
| **Magenta** | End with bb or bd |
| **Box** | Start with *bbbbb* for color group 0, start with *aaaa* for color group 1 |
| **Sphere** | Start with *aa* for color group 1 |
| **Cylinder** | Start with *bbbb* for color group 0, start with *aaa* for color group 1 |
| **Capsule** | Start with *bbb* for color group 0, start with *a* for color group 1 |
| **Ellipsoid** | Start with *bb* for color group 0 |

Table 2: (Top) Potential composition analysis of the messages from round 16,760 (bottom of Table 1). *Italic* symbols are used to specify shapes and roman symbols are used to specify colors. ā indicates deleting a single prefix *a*. (Bottom) Rules for each color and shape derived from the top table.

| - | Box | Sphere | Cylinder | Capsule | Ellipsoid |
|---|---|---|---|---|---|
| **Blue** | *eeeee* e{e,ee} | *eeee* ee | *eeee* e{e,ed} | *eee* e{e,ed} | *ee* e{e,a} |
| **Red** | *eeeee* ēēēē{e,a} | *eeee* ēēēē{e,ea} | *eeee* ēēēē{e,a} | *eee* ēēēē{b,ba} | *ee* ēēēē{a,c} |
| **White** | *bbbb* b | *b* b | *bbb* {b,d} | *bb* {b,d} | *bbb* {d,c} |
| **Gray** | *eeeee* ēēēe | *eeee* ēēē{_,a} | *eeee* ēēē{e,a} | *eee* ēēē{e,a} | *ee* ēēē{b,d} |
| **Yellow** | *bbbb* b̄b̄{c,d} | *b* b̄b̄e{e,ec} | *bbb* b̄b̄{c,d} | *bb* b̄b̄{a,c} | *bbb* b̄b̄{a,e} |
| **Green** | *eeeee* {e,a} | *eeee* {e,a} | *eeee* {e,a} | *eee* {e,a} | *ee* {e,a} |
| **Cyan** | *eeeee* eeea | *eeee* ee{e,a} | *eeee* ee{e,ea} | *eee* ee{e,a} | *ee* eea |
| **Magenta** | *bbbb* b̄{b,d} | *b* b̄{c,d} | *bbb* b̄{b,d} | *bb* b̄b | *bbb* b̄b̄{c,a} |

Table 3: Potential analysis of the messages observed in the zero-shot test. Gray cells indicate object types unseen during the training phase. *Italic* symbols are used to specify shapes and roman symbols to specify colors. b̄ indicates deleting a single prefix *b*. ē indicates deleting a single prefix *e*. Underscores indicate blanks.

even when we consider the effects of b̄ and ē. However, the messages describing the held-out object types show clear structure with the exception of *yellow ellipsoid*.

In order to assess the communication accuracy when held-out objects are involved, we conducted another test with the agents from round 19,980. Each held-out object was given to the speaker, the listener, or both. In the first two cases, the held-out object was paired with all 40 object types and each pair was tested 10 times. In the last case, the held-out object was tested against itself 10 times. In all cases, the agents switched roles after 5 times. Table 4 shows communication accuracies for each case. We can see the agents can successfully communicate most of the time even when given novel objects. The last column shows that the listener is not simply producing 0 to maximize its chance to win the game. It is also notable that the objects described without b̄ or ē show better performance in general. We noticed the communication accuracy for held-out objects seems relatively weak considering the messages used to describe them strongly showed structure. (Table 3). This, however, results from the grammar (*i.e.* structure) being not as straightforward as Table 2, especially with short messages (*i.e.* frequent use of b̄ and ē). The same tendency can be observed for non-held-out objects as described by the per-object communication accuracy Table 9 in Appendix J.

|  | Given to speaker | Given to listener | Given to both |
|---|---|---|---|
| *Blue Box* | 0.97 | 0.97 | 1.00 |
| *Red Sphere* | 0.92 | 0.91 | 0.80 |
| *White Cylinder* | 0.95 | 0.95 | 0.90 |
| *Gray Capsule* | 0.91 | 0.91 | 1.00 |
| *Yellow Ellipsoid* | 0.91 | 0.90 | 1.00 |

Table 4: Communication accuracy when agents were given objects not seen during the training.

From the grammar analysis in the previous section, we have shown that the emerged language strongly follows a well-defined grammar. In the zero-shot test, the agents demonstrated that they can successfully describe novel object, although not perfectly, by also following a similar grammar. Both are, as stated in the beginning of section 3, necessary conditions for any communication to be considered compositional. Therefore we can safely conclude that the emerged language in this work possesses some qualifications to be considered compositional.

## 4 DISCUSSION

In this work, we used the obverter technique to train neural network agents to communicate in a two-person image description game. Through qualitative analysis, visualization and the zero-shot test, we have shown that even though the agents receive raw perception in the form of image pixels, under the right environment pressures, the emerged language had properties consistent with the ones found in compositional languages.

As an evaluation strategy, we followed previous works and focused on assessing the *necessary conditions* of compositional languages. However, the exact definition of compositional language is still somewhat debatable, and, to the best of our knowledge, there is no reliable way to mathematically quantify the degree of compositionality of an arbitrary language. Therefore, in order to encourage active research and discussion among researchers in this domain, we propose for future work, a quantitatively measurable definition of compositionality. We believe compositionality of a language is not binary (*e.g.* language A is compositional/not compositional), but a spectrum. For example, human language has some aspects that are compositional (*e.g.*, syntactic constructions, most morphological combinations) and some that are not (*e.g.*, irregular verb tenses in English, character-level word composition). It is also important to clearly define grounded language and compositional language. If one agent says *abc (eat red apple)* and another says *cba (apple red eat)*, and they both understand each other, are they speaking compositional language? We believe such questions should be asked and addressed to shape the definition of compositionality.

In addition to the definition/evaluation of compositional languages, there are numerous directions of future work. Observing the emergence of a compositional language among more than two agents is an apparent next step. Designing an environment to motivate the agents to disentangle more than two factors is also an interesting direction. Training agents to consider the context (*i.e.* pragmatics), such as giving each agent several images instead of one, is another exciting future work.

### ACKNOWLEDGMENTS

We would like to thank Scott Reed for discussions on pragmatics, Tom Le Paine for advising the visual module architecture, Jakob Foerster for discussions on grammar induction, Sookyung Kim and Joonseok Lee for discussions on human language and compositionality, Phil Blunsom and Jimeng Sun for helpful comments on the manuscript.

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

| sp | hr | ot | pl | |
|----|----|----|----|-----|
| 1 | 0 | 0 | 0 | me |
| 1 | 0 | 0 | 1 | we |
| 1 | 0 | 1 | 1 | mip |
| 0 | 1 | 0 | 0 | you |
| 0 | 1 | 0 | 1 | yall |
| 0 | 1 | 1 | 1 | yup |
| 1 | 1 | 0 | 1 | yumi |
| 0 | 0 | 1 | 0 | one |
| 0 | 0 | 1 | 1 | they |
| 1 | 1 | 1 | 1 | all |

(a) Subject vectors

| values | |
|--------|-----|
| 011001 | happy |
| 011100 | sad |
| 101001 | angry |
| 100011 | tired |
| 110001 | excited |
| 100101 | sick |
| 100110 | hungry |
| 000111 | thirsty |
| 010101 | silly |
| 010011 | scared |

(b) Predicate vectors

| values | |
|--------|-----|
| 0010101001 | one angry |
| 1101010101 | yumi silly |
| 1111100101 | all sick |
| 0111100110 | yup hungry |
| 1011010101 | mip silly |
| 0111100101 | yup sick |
| 0100100011 | you tired |
| 0011000111 | they thirsty |
| 1001011100 | we sad |
| 1000110001 | me excited |

(c) Example meaning vectors

Table 5: Meaning vectors are composed of (a) subject vectors and (b) predicate vectors. (c) shows 10 out of 100 possible meaning vectors.

## A  EMERGENCE OF GRAMMAR, BATALI (1998)

In Batali (1998), the author successfully trained neural agents to develop a structured (*i.e.* grammatical) language using disentangled meaning vectors as the input. Using 10 subject vectors and 10 predicate vectors, all represented as explicit binary vectors, total 100 meaning vectors could be composed(Table 5). Each digit in the subject vector 5a serves a clear role, respectively representing speaker(*sp*), hearer(*hr*), other(*ot*), and plural(*pl*). The predicate vector values, on the other hand, are randomly chosen so that each predicate vector will have three 1's and three 0's. The combination of ten subject vectors and ten predicate vectors allows 100 meaning vectors.

The author used twenty neural agents for the experiment. Each agent was implemented with the vanilla recurrent neural networks (RNN), where the hidden vector $\mathbf{h}$'s size was 10, same as the size of the meaning vector $\mathbf{m}$ in order to treat $\mathbf{h}$ as the agent's *understanding* of $\mathbf{m}$. In each training round a single learner (*i.e.* listener) and ten teachers (*i.e.* speaker) were randomly chosen. Each teacher, given all 100 $\mathbf{m}$'s in random order, generates a message $\mathbf{s}$[5] for each $\mathbf{m}$ and sends it to the learner. The messages are generated using the obverter techinque, which is described in Algorithm 1. The learner is trained to minimize the mean squared error (MSE) between $\mathbf{h}$ (after consuming the $\mathbf{s}$) and $\mathbf{m}$. After the learner has learned from all ten teachers, the next round begins, repeating the process until the error goes below some threshold.

---
**Algorithm 1:** Message generation process used in Batali (1998).

---
1  $\mathbf{h}^{(0)} = \mathbf{0}$ //Initialize RNN hidden layer with zeros;
2  $\mathbf{s} = [\ ]$ //Initialize the message vector;
3  $t = 0$ //Timestep index;
4  $\mathbf{V} = \mathbf{I} \in \mathbb{R}^{4 \times 4}$ //Each row $\mathbf{v}_0, \mathbf{v}_1, \mathbf{v}_2, \mathbf{v}_3$ corresponds to $a, b, c, d$;
5  **while** $|\mathbf{s}| < $ *max message length* **do**
6  $\quad$ $\mathbf{h}_{\mathbf{v}_i}^{(t)} = \sigma(\mathbf{v}_i \mathbf{W}_i + \mathbf{h}^{(t-1)} \mathbf{W}_h + \mathbf{b})$;
7  $\quad$ $i = \mathrm{argmin}_i ||\mathbf{m} - \mathbf{h}_{\mathbf{v}_i}^{(t)}||^2$;
8  $\quad$ $\mathbf{h}^{(t)} = \mathbf{h}_{\mathbf{v}_i}^{(t)}$;
9  $\quad$ Append $i$ to $\mathbf{s}$;
10 $\quad$ **if** $||\mathbf{m} - \mathbf{h}^{(t)}||^2 < threshold$ **then**
11 $\quad\quad$ Terminate;

---

When the training was complete, the author was able to find strong patterns in the messages used by the agents (Table 6). Note that the messages using predicates *tired, scared, sick* and *happy* especially follow a very clear pattern. Batali also conducted a zero-shot test where the agents were trained without the diagonal elements in Table 6 and tested with all 100 meaning vectors. The agents were able to successfully communicate even when held-out meaning vectors were used, but the

---
[5]A message is a sequence of alphabets chosen from *a, b, c* or *d*. The maximum length was set to 20.

| -       | one  | they | you  | yall | yup  | me  | we   | mip  | yumi | all   |
|---------|------|------|------|------|------|-----|------|------|------|-------|
| **tired**   | cda  | cdab | cdc  | cdcb | cdba | cd  | cdd  | cddb | cdcd | cdb   |
| **scared**  | caa  | caab | cac  | cacb | caba | ca  | cad  | cadb | cacd | cab   |
| **sick**    | daa  | daab | dac  | dacb | daba | da  | dad  | dadb | dacd | dab   |
| **happy**   | baa  | baab | bca  | bcab | baac | ba  | badc | bab  | bac  | babc  |
| **sad**     | aba  | abab | ac   | acb  | abac | a   | abdc | abb  | abc  | abbc  |
| **excited** | cba  | cbab | cca  | cacb | cbca | c   | ccdc | cb   | ccb  | cbc   |
| **angry**   | bb   | bbb  | bc   | bcb  | bbc  | b   | bddc | bdb  | bdc  | bdbc  |
| **silly**   | aa   | aaab | aca  | acab | adba | add | addc | adad | adc  | adbc  |
| **thirsty** | dbaa | dbab | dca  | dcba | dbca | dda | ddac | dbad | dcad | dbacd |
| **hungry**  | dbb  | dbbd | dc   | dcb  | dbc  | dd  | ddc  | dbd  | dcd  | dbcd  |

| -       | one  | they | you  | yall | yup  | me  | we   | mip  | yumi | all   |
|---------|------|------|------|------|------|-----|------|------|------|-------|
| **tired**   | *cd*a  | *cd*ab | *cd*c  | *cd*cb | *cd*ba | *cd*  | *cd*d  | *cd*db | *cd*cd | *cd*b   |
| **scared**  | *ca*a  | *ca*ab | *ca*c  | *ca*cb | *ca*ba | *ca*  | *ca*d  | *ca*db | *ca*cd | *ca*b   |
| **sick**    | *da*a  | *da*ab | *da*c  | *da*cb | *da*ba | *da*  | *da*d  | *da*db | *da*cd | *da*b   |
| **happy**   | *ba*a  | *ba*ab | *bca*  | *bca*b | *ba*ac | *ba*  | *ba*dc | *ba*b  | *ba*c  | *ba*bc  |
| **sad**     | *a*ba  | *a*bab | *a*c   | *a*cb  | *a*bac | *a*   | *a*bdc | *a*bb  | *a*bc  | *a*bbc  |
| **excited** | *c*ba  | *c*bab | *c*ca  | *c*acb | *c*bca | *c*   | *c*cdc | *c*b   | *c*cb  | *c*bc   |
| **angry**   | *c*b   | *c*bb  | *c*c   | *c*cb  | *c*bc  | *c*   | *c*ddc | *c*db  | *c*dc  | *c*dbc  |
| **silly**   | (aa)   | (aaab) | (aca)  | (acab) | *ad*ba | *ad*d | *ad*dc | *ad*ad | *ad*c  | *ad*bc  |
| **thirsty** | *db*aa | *db*ab | *dc*a  | *dc*ba | *db*ca | *dd*a | *dd*ac | *db*ad | *dc*ad | *db*acd |
| **hungry**  | *db*b  | *db*bd | *dc*   | *dc*b  | *db*c  | *dd*  | *dd*c  | *db*d  | *dc*d  | *db*cd  |

Table 6: (Top) Messages used by a majority of the population for each of the given meanings. (Bottom) A potential analysis of the system in terms of a root plus modifications. *Italic* symbols are used to specify predicates and roman symbols are used to specify subjects. Messages in parentheses cannot be made to fit into this analysis.

messages used for the held-out meaning vectors did not show as strong compositional patterns as the non-zero-shot case.

## B  COMPARISON BETWEEN THE OBVERTER TECHINQUE AND THE REINFORCEMENT LEARNING-BASED APPROACH

The obverter technique allows us to generate messages that encourage the agents to use a shared language, even a highly structured one, via using a single RNN for both speaking and listening.

This is quite different from other RL-based related works (Lazaridou et al., 2017; Mordatch & Abbeel, 2017; Foerster et al., 2016; Jorge et al., 2016; Kottur et al., 2017) where each agent has separate components (*e.g.* two RNNs) for generating messages and consuming messages. This is necessary typically because the message generation module and the message consumption module have different input/output requirements. The message generation module accepts some input related to the task (*e.g.* goal description vector, question embedding, or image embedding) and generates discrete symbols. The message consumption module, on the other hand, accepts discrete symbols (*i.e.* the message) and generates some output related to the task (*e.g.* some prediction or some action to take). Therefore, when a neural agent speaks in the RL-based approach, its message generation process is completely separated from its own listening process, but tied to the listening process of another agent (*i.e.* listener)[6]. This means an agent may not have internal consistency; what an agent speaks may not make sense to itself. However, agents in the RL-based setting do converge on a common language because, during the training, the error signal flows from the listener to the speaker directly.

Obverter approach, on the other hand, requires that each agent has a single component for both message generation and message consumption. This single component accepts discrete symbols and generates some output related to the task. This guarantees internal consistency because an agent's message generation process is tied to its own message consumption process; it will only generate

---

[6]Note that this is still true even when using tricks to replace the message sampling process with some approximation functions (Gumbel-softmax or Concrete distribution)

messages that make sense to itself. In the obverter setting, the error signal does not flow between agents directly, but agents converge on a common language by taking turns to be the listener; the listener tries to understand what the speaker says, so that when the listener becomes the speaker, it can generate messages that make sense to itself and, at the same time, will be understood by the former speaker (now listener).

The advantage of obverter approach over RL-based approach is that it is motivated by the theory of mind and more resembles the acquisition/development process of human language. Having a single mechanism for both speaking and listening, and training oneself to be a good listener leads to the emergence of self-consistent, shared language. However, obverter technique requires that all agents perform the same task, which means all agents must have identical model architectures. This is because, during the message generation process, the speaker internally simulates what the listener will go though when it hears the message. Therefore we cannot play an asymmetrical game such as where the speaker sees only one image and generates a message but the listener is given multiple images and must choose one after hearing the message. RL-based approaches do not have this problem since there are separate modules for speaking and listening.

We believe obverter technique could be the better choice for certain tasks regarding human mind emulation. But it certainly is not the tool for every occasion. The RL-based approach is a robust tool for any general task that may or may not involve human-like communication. We conclude this section with a possible future research direction that combines the strengths of both approaches to enable communication in more interesting and complicated tasks.

## C  MODEL ARCHITECTURE DETAILS

We used TensorFlow and the Sonnet library for all implementation.

### C.1  VISUAL MODULE

We used an eight-layer convolutional neural network. We used 32 filters with the kernel size 3 for every layer. The strides were $[2, 1, 1, 2, 1, 2, 1, 2]$ for each layer. We used rectified linear unit (ReLU) as the activation function for every layer. Batch normalization was used for every layer. We did not use the bias parameters since we used Batch normalization. For padding, we used the TensorFlow VALID padding option for every layer. The fully connected layer that follows the convolutional neural network was of 256 dimensions, with ReLU as the activation function.

### C.2  LANGUAGE MODULE

We used a single layer Gated Recurrent Units (GRU) to implement the language module. The size of the hidden layer was 64.

### C.3  DECISION MODULE

We used a two-layer feedforward neural network. The first layer reduces the dimensionality to 128 with ReLU as the activation function, then the second layer generates a scalar value with sigmoid as the activation function.

## D  MESSAGE GENERATION ALGORITHM USED IN OUR WORK

## E  TRAINING DETAILS

Both agents' model parameters are randomly initialized. The training process consists of rounds where teacher/learner roles are changed, and each round consists of multiple games where learner's model parameters are updated. In each game, the teacher, given a mini-batch of images, generates corresponding messages. The learner, given a separate mini-batch of images and the messages from the teacher, decides whether it is seeing the same object type as the teacher. Learner's model parameters are updated to minimize the cross entropy loss. After playing a predefined number of games, we move on to the next round where two agents change their roles.

---

**Algorithm 2:** Message generation process used in our work.

1   $\mathbf{h}^{(0)} = \mathbf{0}$ //Initialize GRU hidden layer with zeros;
2   $\mathbf{s} = [\ ]$ //Initialize the message vector;
3   $t = 0$ //Timestep index;
4   $\mathbf{V} = \mathbf{I} \in \mathbb{R}^{5 \times 5}$ //Each row $\mathbf{v}_0, \mathbf{v}_1, \mathbf{v}_2, \mathbf{v}_3, \mathbf{v}_4$ corresponds to $a, b, c, d, e$;
5   $\mathbf{x} = image$;
6   $\mathbf{z} = VisualModule(\mathbf{x})$;
7   **while** $|\mathbf{s}| < max\ message\ length$ **do**
8      $\mathbf{h}_{\mathbf{v}_i}^{(t)} = GRU(\mathbf{v}_i, \mathbf{h}^{(t-1)})$;
9      $i = \text{argmax}_i\ DecisionModule([\mathbf{h}_{\mathbf{v}_i}^{(t)}, \mathbf{z}])$;
10      $\mathbf{h}^{(t)} = \mathbf{h}_{\mathbf{v}_i}^{(t)}$;
11      Append $i$ to $\mathbf{s}$;
12      **if** $DecisionModule([\mathbf{h}_{\mathbf{v}_i}^{(t)}, \mathbf{z}]) > threshold$ **then**
13         Terminate;

---

We found twenty games per round, with fifty images per mini-batch to work well. We repeat the rounds for $20,000$ times. Further rounds did not improve the results, or even degraded the performance. For vocabulary size (*i.e.* number of unique symbols) and the maximum message length, we used 5 and 20 respectively, similar to what Batali (1998) used. Note that when generating a message using the obverter technique, the generation process stops as soon as the speaker's (*i.e.* teacher) output $\hat{y}$ becomes bigger than some threshold. In our work, we experimented with various values from $0.5$ to $0.95$, and found higher values to work better than lower values. We used $0.95$ for all our final experiments.

## F   COMMUNICATION ACCURACY FOR EACH OBJECT TYPE

| - | Box | Sphere | Cylinder | Capsule | Ellipsoid |
|---|---|---|---|---|---|
| **Blue** | 97.75 | 97.00 | 95.00 | 93.75 | 93.50 |
| **Red** | 95.50 | 93.00 | 95.75 | 95.25 | 97.00 |
| **White** | 95.25 | 96.25 | 93.50 | 94.50 | 97.00 |
| **Gray** | 93.00 | 95.00 | 94.25 | 96.75 | 97.25 |
| **Yellow** | 98.00 | 95.00 | 95.50 | 94.25 | 94.25 |
| **Green** | 96.00 | 93.50 | 95.00 | 94.50 | 95.25 |
| **Cyan** | 97.50 | 94.50 | 97.00 | 94.00 | 94.75 |
| **Magenta** | 95.25 | 96.75 | 94.75 | 94.50 | 95.25 |

Table 7: Accuracy when each object type is given to the speaker.

We conducted a separate test with the agents from round $16,760$ to assess the communication accuracy for each object type. The agents were given $1,600$ total object pairs ($40 \times 40$). Each object pair was tested 10 times, where after 5 times the agents switched speaker/listener roles. The average accuracy was $95.4\%$, and only 88 out of $1,600$ object pairs were communicated with accuracy lower than $0.8$.

Table 7 describes the accuracy when each object type was given to the speaker. We can observe that the accuracy is higher for objects that are described with less overlapping messages. For example, *yellow box* is communicated with the accuracy of $98\%$, and it is described with *aaaaaa*, which is not used for any other object types. *Gray box*, on the other hand, is communicated with accuracy $93\%$. It is described with *aaa*, which is also used for *yellow capsule* and *green sphere*, both of which are communicated with low accuracies as well.

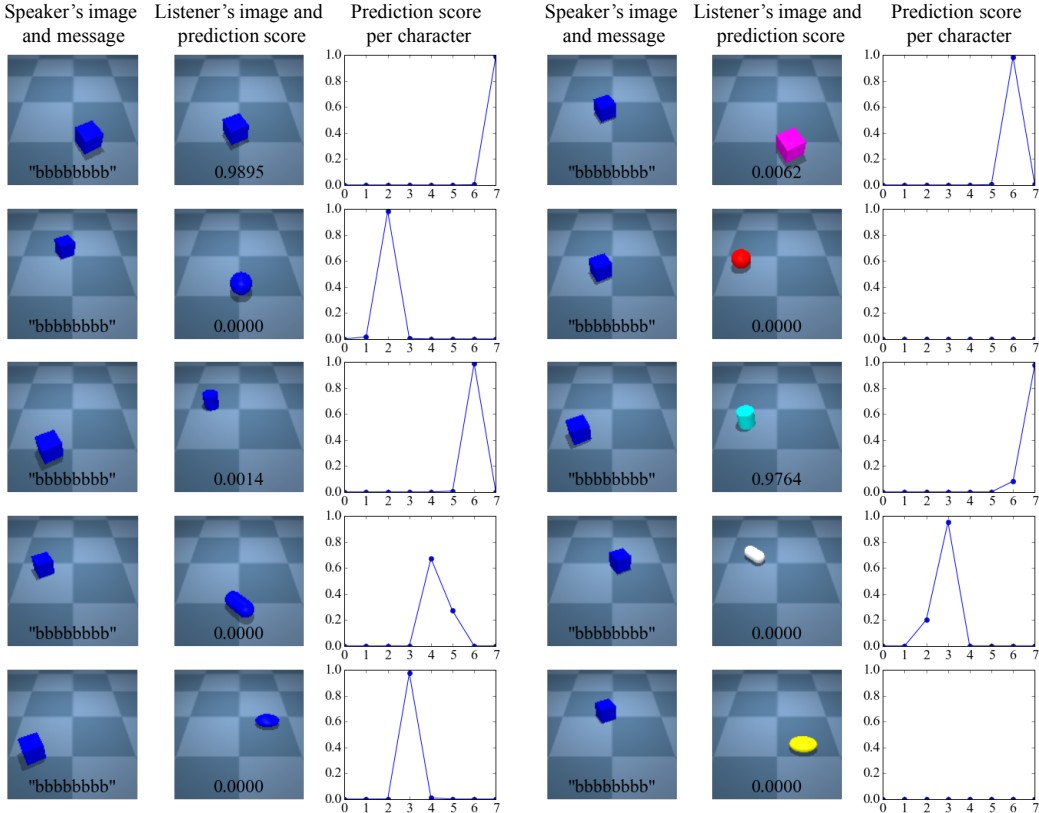

Figure 5: Ten communication examples when the speaker is given a blue box. Five examples on the left show when the listener is given blue objects. Five examples on the right show when the listener is given objects of different colors.

## G    COMMUNICATION EXAMPLE AND ANALYSIS

Figure 5 provides ten examples of communication when the speaker is given a *blue box* and the listener is given various object types. The listener's belief (*i.e.* score) that it is seeing the same image as the speaker changes each time it consumes a symbol. It is notable that most of the time the score jumps between 0 and 1, rather than gradually changing in between. This is natural given that messages that differ by only a single character can mean different objects (*e.g. blue box* and *blue cylinder*). This phenomenon can also be seen in human language. For example, *blue can* and *blue cat* differ by a single alphabet, but the semantics are completely different.

Object types that are described by similar messages as *blue box*, such as *blue cylinder* and *magenta box* cause marginal confusion to the listener such that prediction scores for both objects are not complete zeros. There are also cases where two completely different objects are described by the same message as mentioned in Section 3.1. From Table 2 we can see that *blue box* and *cyan cylinder* are described by the same message *bbbbbbb{b,d}*, although the messages were composed using different rules. Therefore the listener generates high scores for both objects, occasionally losing the game when the agents are given this specific object pair (1 out of 40 chance). This can be seen as a side effect coming from the *principle of least effort* which motivates the agents to win the game most of the time while minimizing the effort to generate messages.

## H    VISUALIZATION OF IMAGE EMBEDDINGS

Section 3.2 provides strong evidence that the agents are properly recognizing the color and shape of an object. In this section, we study the visual module of both agents to study how they are processing

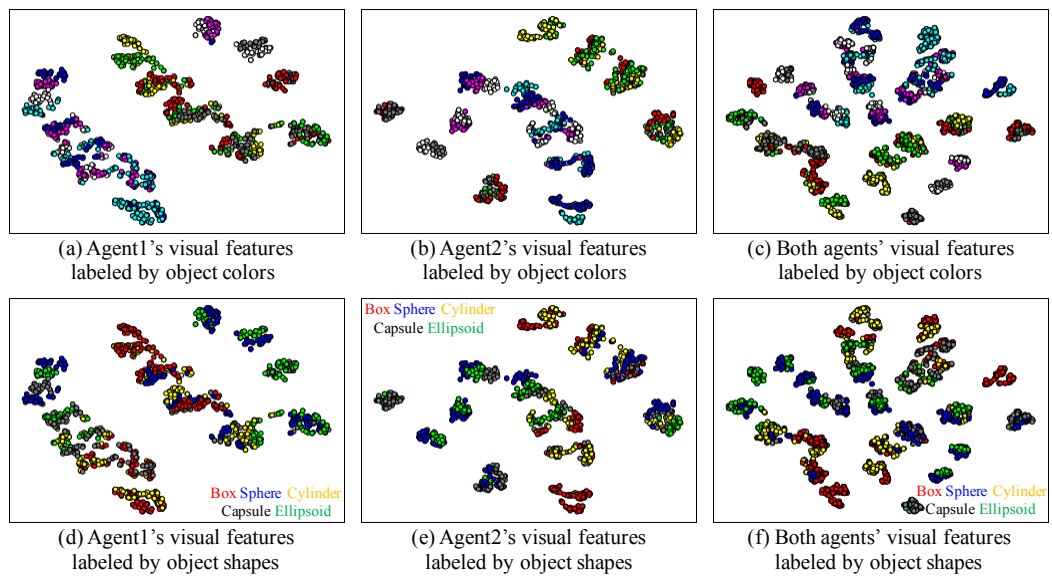

Figure 6: Scatter plots of the image embedding from the agents' visual module. T-SNE was used to reduce the dimension to 2D.

the pixel input. We give each agent 1600 images, 40 per object type, and take their image embeddings (output of the fully-connected layer in the visual module). We use t-SNE (Maaten & Hinton, 2008) to reduce the dimensionality to 2D, and generate scatter plots as shown by Figure 6. The top row and the bottom row are the same scatter plots, but the dots are colored with different labels; the top row shows the color of each object, and the bottom row shows the shape of each object.

It is notable that the image embeddings form clear clusters, and all clusters are quite disentangled from one another. This means that the agents have learned to differentiate objects by their color and shape, and each color and shape have a specific place in the agents' mind. It is also impressive that not only both agents learned similar relationships between colors and shapes(Figure 6 (a) and (b) show similar clusters, as do (d) and (e)), but also they learned similar absolute values for colors and shapes. Even when we plot image embeddings from both agents together (Figure 6 (c) and (f)) the cluster qualities are kept almost identical (with slightly higher number of clusters) to when we plot them separately. Therefore when one agent thinks of color *red*, and utters a message to describe it, the other agent hears the message and think of something *red* as well. This is, of course, what we wanted to achieve by using the obverter technique.

The fact that a couple of color or shapes are occasionally in the same cluster suggests the agents have not perfectly disentangled colors and shapes. For perfect disentanglement, we believe some modifications to the image description game is required, which can be an important topic for future work. However, studying the cluster sheds some light on why the agents generate specific messages for each object type. For example, in Figure 6 (a) and (b), *blue, white, cyan* and *magenta* are often intertwined or their respective clusters are located nearby, as is the case for *red, gray, yellow* and *green*. This suggests the reason agents use prefix *b*'s to specify shapes for former color group and prefix *a*'s to specify shapes for the latter color group. Additionally, in Figure 6 (d) and (e), *Box* and *cylinder* are often located nearby, and *sphere* and *ellipsoid* show similar behavior[7]. We conjecture that this is the reason the messages describing *box* and *cylinder* are similarly long, and the messages describing *sphere* and *ellipsoid* tend to be similarly short.

---

[7]Incidentally, boxes and cylinders are of similar shape, which is also the case for spheres and ellipsoids.

## I  ORIGINAL MESSAGES FROM THE ZERO-SHOT TEST

| - | Box | Sphere | Cylinder | Capsule | Ellipsoid |
|---|-----|--------|----------|---------|-----------|
| **Blue** | eeeeeee{e,ee} | eeeeee | eeeeee{e,ed} | eeee{e,ed} | eee{e,a} |
| **Red** | ee{e,a} | e,ea | e{e,a} | b,ba | a,c |
| **White** | bbbbb | bb | bbb{b,d} | bb{b,d} | bbb{d,c} |
| **Gray** | eeee | e,ea | ee{e,a} | e,a | b,d |
| **Yellow** | bb{c,d} | e{e,ec} | b{c,d} | a,c | a,e |
| **Green** | eeeee{e,a} | eeee{e,a} | eeeee{e,a} | eee{e,a} | ee{e,a} |
| **Cyan** | eeeeeeeeea,a×20 | eeeeee{e,a} | eeeeeee{e,ea} | eeeee{e,a} | eeeea |
| **Magenta** | bbb{b,d} | c,d | bb{b,d} | bb | b{c,a} |

Table 8: Messages most often used by the two agents when speaking about a given object. Shaded cells indicate the objects not seen during the training. Brackets indicate the variation often seen at the last character.

## J  COMMUNICATION ACCURACY FOR EACH OBJECT TYPE IN ZERO-SHOT TEST

| - | Box | Sphere | Cylinder | Capsule | Ellipsoid |
|---|-----|--------|----------|---------|-----------|
| **Blue** | 96.75 | 95.00 | 94.75 | 95.75 | 96.50 |
| **Red** | 94.25 | 90.00 | 94.25 | 96.75 | 91.25 |
| **White** | 99.00 | 96.00 | 95.75 | 97.25 | 95.75 |
| **Gray** | 93.50 | 91.00 | 94.75 | 91.50 | 96.00 |
| **Yellow** | 98.75 | 94.75 | 95.75 | 92.00 | 91.50 |
| **Green** | 95.00 | 95.00 | 95.00 | 95.00 | 94.50 |
| **Cyan** | 92.00 | 95.00 | 97.50 | 95.00 | 95.00 |
| **Magenta** | 95.25 | 96.00 | 97.00 | 96.00 | 95.75 |

Table 9: Accuracy when each object type is given to the speaker. Shaded cells indicate the objects not seen during the training.

In the same manner as Appendix F, we conducted a separate test with the agents from round $19,980$ to assess the communication accuracy for each object type. The agents were given $1,600$ total object pairs ($40 \times 40$). Each object pair was tested $10$ times, where after $5$ times the agents switched speaker/listener roles. The average accuracy was $94.73\%$, and $103$ out of $1,600$ object pairs were communicated with accuracy lower than $0.8$.

Table 9 describes the accuracy when each object type was given the speaker. Shaded cells indicate objects not seen during the training. Here we can observe the same tendency as the one seen in Appendix F; the accuracy is higher for objects that are described with less overlapping messages.

## K  AN EXAMPLE OF USING NEGATION TO PASS A ZERO-SHOT TEST

Lets assume agent0 is aware of red circle, blue square and green triangle. If agent0 came upon a blue circle for the first time and had to describe it to agent1, the efficient way would be to say blue circle. But it could also say blue not_square not_triangle. If agent1 had a similar knowledge as agent0 did, then both agents would have a successful communication. However, it is debatable whether saying blue not_square not_triangle is as compositional as blue circle.

