# OpenReview forum: "Compositional Obverter Communication Learning from Raw Visual Input"
_ICLR.cc/2018/Conference — Accept (Poster)_

### Official Review · AnonReviewer2 · 2017-11-19
**very interesting paper, more details would help**

**Rating:** 9
**Confidence:** 4

**Review:**

This paper proposes to apply the obverter technique of Batali (1998) to a multi-agent communication game. The main novelty with respect to Batali's orginal work is that the agents in this paper have to communicate about images represented at the raw pixel level. The paper presents an extensive analysis of the patterns learnt by the agents, in particular in terms of how compositional they are.

I greatly enjoyed reading the paper: while the obverter idea is not new, it's nice to see it applied in the context of modern deep learning architectures, and the analysis of the results is very interesting.

The writeup is somewhat confusing, and in particular the reader has to constantly refer to the supplementary materials to make sense of the models and experiments. At least the model architecture could be discussed in more detail in the main text.

It's a pity that there is no direct quantitative comparison between the obverter technique and a RL architecture.

Some more detailed comments:

It would be good if a native speaker could edit the paper for language and style (I only annotated English problems in the abstract, since there were just too many of them).

Abstract:

distinguishing natures -> aspects

describe the complex environment -> describe complex environments

with the accuracy -> with accuracy

Introduction:

have shown great progress -> has...

The claim that language learning in humans is entirely based on communication needs hedging. See, e.g., cultures where children must acquire quite a bit of their language skills from passive listening to adult conversations (Schieffelin & Ochs 1983; Shore 1997).

Also, while it's certainly the case that humans do not learn from the neatly hand-crafted features favored in early language emergence studies, their input is most definitely not pixels (and, arguably, it is something more abstract than raw visual stimuli, see, e.g., the work on "object biases" in language acquisition).

Method:

The description of the experiment sometimes refers to the listener having to determine whether it is seeing the same image as the speaker, whereas the game consists in the listener telling whether it sees the same object as the speaker. This is important, since, crucially, the images can present the same object in different locations.

Section 2.2 is very confusing, since the obverter technique has not been introduced, yet, so I kept wondering why you were only describing the listener architecture. Perhaps, current section 2.3 should be moved before 2.2.

Also in 2.2: what is the "meaning vector" that BAtali's RNN hidden vector has to be close to?

It's confusing to refer to the binary 0 and 1 vectors as outputs of the sigmoid: they are rather the binary categories you want the agents to approximate with the sigmoid function.

The obverter technique should be described in more detail in the main text. In particular, with max message length 20 and a vocabulary of 5 symbols, you'll have an astronomical number of possible inputs to evaluate at each step: is this really what you're doing?

Experiments:

distinctness: better call it distinctiveness

Is training accuracy in Fig 4 averaged across both agents? And what were the values the Jaccard similarity was computed over?

It's nice that the agents discovered some non-strictly agglutinative morphology, consisting in removing symbols from the prefix. Also, it looks like they are developing some notion of "inflectional class" (Aronoff 1994) (the color groups), which is also intriguing. However, I did not understand rules such as: "remove two as and add one a"... isn't that the same as: "add one a"?

Discussion and future work:

I didn't follow the discussion about partial truth and using negations.

There is another ICLR submission that proposes a quantitative measure of compositionality you might find interesting: https://openreview.net/pdf?id=HJGv1Z-AW

---

> ### Author Response · Authors · 2018-01-04
> **Authors' response to the reviewers' comments.**
>
> We thank you for your review and constructive comments. We address your concerns as follows.
>
> 1. The claim that language learning in humans is entirely based on communication needs hedging:
> We appreciate the suggested reading. We revised our paper to be more appropriate in our claims.
>
> 2. Human language acquisition is based on somewhere between hand-engineered features and pixels:
> Your point is true, and it led us re-check the contribution of our paper. The agents in our work learns to develop both the linguistic ability and visual cognition simultaneously. The CNN and the RNN of the agents are initialized with random parameters. Through training, CNN is updated to disentangle shapes and colors, while RNN is updated to describe the shapes and colors. Therefore, similar to your point, after the CNN is updated to a certain level, RNN no longer deals with completely raw signals. However, at the early training rounds, the RNN does deal with seemingly random values (since the CNN has not learned much yet) and maybe that is why it takes several thousand training rounds for the agents to show some performance gain. We appreciate this insightful comment, and we revised the paper to incorporate this comment.
>
> 3. The description of the game confusingly uses the word “image” and “object”:
> We revised the paper to be more consistent in using those two words.
>
> 4. Maybe section 2.3 should come before section 2.2:
> In the revised paper, we mention in section 2.2 that the obverter strategy will be described in detail in section 2.3.
>
> 5. What is the meaning vector in Batali’s work?
> We revised the paper so that there is a brief description of the meaning vector, and we also inform the readers to refer to the supplementary material for detailed information.
>
> 6. The output of the agents are not vectors but sigmoid values:
> We revised the paper to remove the confusion.
>
> 7. The obverter technique should be described in more detail:
> We revised the paper to provide more detailed description of the obverter strategy. Also, we explain that the characters are chosen greedily at each timestep. We also discuss the deterministic nature of the obverter strategy in the revised paper (please see our second explanation for Reviewer 3 for more detail)
>
> 8. Is training accuracy in Fig 4 averaged across both agents?
> In a single training round, only one agent (i.e. the learner) makes the prediction. We aggregated those predictions to calculate the accuracy. Therefore, in round 0, agent 0’s prediction accuracy is taken. In round 1, agent 1’s prediction accuracy is taken. And we repeat this process. This is described this in line 146 by saying “Training accuracy was calculated by rounding the learner’s sigmoid output by 0.5”.
>
> 9. What were the values the Jaccard similarity was computed over?
> If agent0 generates “aaa”, “aba”, “abb” to describe red box, and agent1 generates “aaa”, “aba”, “abc” to describe red box, then we calculate the Jaccard similarity by number_of_duplicate_messages/number_of_unique_messages, which is 2/4=0.5. We do this for all object types (total 40) and average the Jaccard similarity values to obtain the final value.
>
> 10. However, I did not understand rules such as: "remove two as and add one a"... isn't that the same as: "add one a"?
> The suffix used to describe the color gray is “a_hat, a_hat, a” The two ‘a_hat’s indicate removing two ‘a’s from the prefix (which is responsible for describing shapes). For example, gray box uses “aaaa” as the prefix and “a_hat a_hat a” as the suffix. Therefore putting them together gives us “aaa”, which is the message that describes gray box as shown in the bottom of Table 1. This is different from adding a single ‘a’ to the prefix “aaaa”, which gives us “aaaaa”.
> In the revised paper, we made it clear that a_hat is for deleting the prefix.
>
> 11. I didn't follow the discussion about partial truth and using negations:
> Let’s assume I am aware of red circle, blue square and green triangle. If I came upon a blue circle for the first time and had to describe it to the listener, I could say “blue circle”. But I could also say “blue not_square not_triangle”. If the listener had a similar knowledge as I did, then we would have a successful communication. However, it is debatable whether saying “blue not_square not_triangle” is as compositional as “blue circle”.
> We provide this explanation in the supplementary material in the revised paper.
>
> 12. Another ICLR submission a quantitative measure of compositionality:
> Thank you for the pointer, we will look into this.

---

> > ### Comment · AnonReviewer2 · 2018-01-05
> > **thanks for the clarifications**
> >
> > Thanks for your clarifications. I generally disagree with AnonReviewer1 negative opinion, however it would help if you did define more clearly what you mean for Compositionality.

---

> > > ### Author Response · Authors · 2018-01-05
> > > **Thank you for the response**
> > >
> > > We thank you for your response.
> > > Based on AnonReviewer1's comment, we also decided that providing a more explicit definition of compositionality would improve the paper. As we wrote in our second item to AnonReviewer1's review, we revised the paper to incorporate this point. Specifically as follows:
> > > - In the second paragraph of Introduction in the revised manuscript, we provide an explicit definition of compositionality.
> > > - In the last paragraph of page 7 of the revised manuscript, we link the definition of compositionality to the experiment results.

---

### Official Review · AnonReviewer1 · 2017-11-25

**Rating:** 3
**Confidence:** 4

**Review:**

This paper presents a technique for training a two-agent system to play a simple reference game involving recognition of synthetic images of a single object.  Each agent is represented by an RNN that consumes an image representation and sequence of tokens as input, and generates a binary decision as output. The two agents are initialized independently and randomly. In each round of training, one agent is selected to be the speaker and the other to be the listener. The speaker generates outputs by greedily selecting a sequence of tokens to maximize the probability of a correct recognition w/r/t the speaker's model. The listener then consumes these tokens, makes a classification decision, incurs a loss, and updates its parameters. Experiments find that after training, the two agents converge to approximately the same language, that this language contains some regularities, and that agents are able to successfully generalize to novel combinations of properties not observed during training.

While Table 3 is suggestive, this paper has many serious problems. There isn't an engineering contribution---despite the motivation at the beginning, there's no attempt to demonstrate that this technique could be used either to help comprehension of natural language or to improve over the numerous existing techniques for automatically learning communication protocols.  But this also isn't science: "generalization" is not the same thing as compositionality, and there's no testable hypothesis articulated about what it would mean for a language to be compositional---just the post-hoc analysis offered in Tables 2 & 3. I also have some concerns about the experiment in Section 3.3 and the overall positioning of the paper.

I want to emphasize that these results are cool, and something interesting might be going on here! But the paper is not ready to be published. I apologize in advance for the length of this review; I hope it provides some useful feedback about how future versions of this work might be made more rigorous.

WHAT IS COMPOSITIONALITY?

The title, introduction, and first three sections of this paper emphasize heavily the extent to which this work focuses on discovering "compositional" language. However, the paper doesn't even attempt to define what is meant by compositionality until the penultimate page, where it asserts that the ability to "accurately describe an object [...] not seen before" is "one of the marks of compositional language". Various qualitative claims are made that model outputs "seem to be" compositional or "have the strong flavor of" compositionality. Finally, the conclusion notes that "the exact definition of compositional language is somewhat debatable, and, to the best of our knowledge, there was no reliable way to check for the compositionality of an arbitrary language."

This is very bad.

It is true that there is not a universally agreed-upon definition of compositionality. In my experience, however, most people who study these issues do not (contra the citation-less 5th sentence of section 4) think it is simply an unstructured capacity for generalization.  And the implication that nobody else has ever attempted to provide a falsifiable criterion, or that this paper is exempt from itself articulating such a criterion, is totally unacceptable.  (You cannot put "giving the reader the tools to evaluate your current claims" in future work!)

If this paper wishes to make any claims about compositionality, it must _at a minimum_:

1. Describe a test for compositionality.

2. Describe in detail the relationship between the proposed test and other definitions of compositionality that exist in the literature.

3. If this compositionality is claimed to be "language-like", extend and evaluate the definition of compositionality to more complex concepts than conjunctions of two predicates.

Some thoughts to get you started: when talking about string-valued things, compositionality almost certainly needs to say something about _syntax_. Any definition you choose will be maximally convincing if it can predict _without running the model_ what strings will appear in the gray boxes in Figure 3.  Similarly if it can consistently generate analyses across multiple restarts of the training run.  The fact that analysis relies on seemingly arbitrary decisions to ignore certain tokens is a warning sign. The phenomenon where every color has 2--3 different names depending on the shape it's paired with would generally be called "non-compositional" if it appeared in a natural language.

This SEP article has a nice overview and bibliography: https://plato.stanford.edu/entries/compositionality/. But seriously, please, talk to a linguist.

MODEL

The fact that the interpretation model greedily chooses symbols until it reaches a certain confidence threshold would seem to strongly bias the model towards learning a specific communication strategy. At the same time, it's not actually possible to guarantee that there is a greedily-discoverable sequence that ever reaches the threshold! This fact doesn't seem to be addressed.

This approach also completely rules out normal natural language phenomena (consider "I know Mary" vs "I know Mary will be happy to see you"). It is at least worth discussing these limitations, and would be even more helpful to show results for other architectures (e.g. fixed-length codes or loss functions with an insertion penalty) as well.

There's some language in the appendix ("We noticed that a larger vocabulary and a longer message length helped the agents achieve a high communication accuracy more easily. But the resulting messages were challenging to analyze for compositional patterns.") that suggests that even the vague structure observed is hard to elicit, and that the high-level claim made in this paper is  less robust than the body suggests. It's _really_ not OK to bury this kind of information in the supplementary material, since it bears directly on your core claim that compositional structure does arise in practice. If the emergence of compositionality is sensitive to vocab size & message length, experiments demonstrating this sensitivity belong front-and-center in the paper.

EVALUATION

The obvious null hypothesis here is that unseen concepts are associated with an arbitrary (i.e. non-compositional) description, and that to succeed here it's enough to recognize this description as _different_ without understanding anything about its structure. So while this evaluation is obviously necessary, I don't think it's powerful enough to answer the question that you've posed. It would be helpful to provide some baselines for reference: if I understand correctly, guessing "0" identically gives 88% accuracy for the first two columns of Table 4, and guessing based on only one attribute gives 94%, which makes some of the numbers a little less impressive.

Perhaps more problematically, these experiments don't rule out the possibility that the model always guesses "1" for unseen objects. It would be most informative to hold out multiple attributes for each held-out color (& vice-versa), and evaluate only with speakers / listeners shown different objects from the held-out set.

POSITIONING AND MOTIVATION

The first sentence of this paper asserts that artificial general intelligence requires the ability to communicate with humans using natural language. This paper has nothing to do with AGI, humans, or human language; to be blunt, this kind of positioning is at best inappropriate and at worst irresponsible. It must be removed. For the assertion that "natural language processing has shown great progress", the paper provides a list of citations employing neural networks exclusively and beginning in 2014 (!). I would gently remind the authors that NLP research did not begin with deep learning, and that there might be slightly earlier evidence for their claim.

The attempt to cite relevant work in philosophy and psychology is commendable!  However, many of these citations are problematic, and some psycho-/historico-linguistic claims are still missing citations. A few examples: Ludwig Wittgenstein died in 1951, so it is somewhat surprising to see him cited for a 2010 publication (PI appeared in 1953); similarly Zipf (2016).  The application of this Zipf citation is dubious; the sentence preceded by footnote 7 is false and has nothing to do with the processes underlying homophony in natural languages. I would encourage you to consult with colleagues in the relevant fields.

---

> ### Author Response · Authors · 2018-01-04
> **Authors' response to the reviewers' comments. (part 2)**
>
>
> - Greedily choosing the characters has limits
> Please see our second explanation for Reviewer 3.
>
> - Hyperparameters (vocabulary size, maximum length of the message) should not be in the supplementary material.
> Thank you for pointing this out. We agree that they are more appropriately discussed in section 2.4 where we discuss the environmental pressure. This is reflected in the revised paper.
>
> - Table 4 is not sufficient to confirm compositionality:
> We described this in line 244 (section 4). We agree that currently many works in this field rely on evaluations that check the necessary conditions of compositional language, and we lack ones that check the sufficient conditions of compositional language. Although, we are not sure if there are such measures. It is another line of future work for all of us.
>
> - Table 4. Always guessing 0 gives 88% accuracy:
> To be precise, always guessing 0 gives 97.5% accuracy. Because out of 40 object pairs, you would only be wrong once (when the same object type is paired). However, as the numbers in Table 4 shows, the agents do not always guess 0.
>
> - Table 4. Guessing based on only one attribute gives 94% accuracy:
> If the agents only focus on colors (which is the dominant attribute, since there are 8 colors and 5 shapes), then out of 40 object pairs, they would be wrong 4 times (when the object pair consists of the same color but different shapes). So that gives us 90% accuracy. However, Table 4 shows that the agents are doing better than 90% most of the time, telling us that they are not just focusing on one attribute.
>
> - Maybe the agents always guess 1 for unseen objects.
> If they did, then the last column of Table 4 should be all 1.0.
>
> - Saying artificial intelligence requires the ability to communicate with humans is problematic:
> It’s hard to understand why that statement is problematic. We already have simple AI agents around us that try to communicate with us, such as Google Home, Siri, and Alexa. It is hard to imagine that, in the future, we will not verbally communicate with AI.
>
> - NLP has longer history than deep learning:
> Yes, it is true. We agree that our citations were biased due to the venue we are submitting this work to, ICLR, which was born recently after the renaissance of neural networks. We incorporated this point in the revised paper.
>
> - The sentence preceded by footnote 7 is false and has nothing to do with the processes underlying homophony in natural languages:
> We can generally observe that homonyms are seldom used in the same context (e.g. mean1(definition: being rude) and mean2(definition: average)), as that will create confusion, leading to inefficient communication. The same could be observed in the emergent language as described in the second paragraph of section G.
> However, we do agree that our logic for connecting the principle of least effort with homonyms might be a bit of a stretch. We removed the sentence in the revised paper.

---

> > ### Comment · AnonReviewer1 · 2018-01-11
> > **Response to rebuttal**
> >
> > Dear authors,
> >
> > Thank you very much for the detailed response. I've spent a while thinking about this, and my score stays the same. 3 points:
> >
> > 1. It is claimed that "the messages in the gray boxes of Figure 3 do actually follow the patterns nicely". I disagree. The central problem is that the paper provides no standard by which to adjudicate our disagreement. If the intention of this paper "was not to develop a rigorous, quantifiable definition of compositionality and its evaluation strategy", then it should not have made any claims that require such a definition; at present these claims are the main attraction.
> >
> > 2. Re: Table 4. You're right about the 88 -> 97.5% (I was lumping the held-out objects together). My 94% was based on imagining that agents predict as normal for objects seen at training time, and only guess based on one attribute for held-out objects. For the identically-1 possibility, the current version of the paper doesn't rule out the possibility that this is the model's strategy on some subset of the held-out objects. Again, I think these are serious issues for the core claim in the paper.
> >
> > 3. Re: citations. It appears the complaint that this paper doesn't cite any pre-deep-learning work has been addressed by acknowledging that earlier work exists, and continuing not to cite it. You can do better.

---

> ### Author Response · Authors · 2018-01-04
> **Authors' response to the reviewers' comments. (part 1)**
>
> We thank you for your detailed review and helpful comments. We address your concerns as follows.
>
> - Lack of engineering contribution:
> Please see our third explanation for Reviewer3.
>
> - The paper doesn't even attempt to define what is meant by compositionality until the penultimate page:
> Although we provided the definition of compositionality in line 33-34, we do agree that the definition could be more explicit. In the revised paper, we explicitly provide the definition of compositionality (or at least what people generally agree to be the definition), and connect that definition to the experiment section to claim that our resulting language does meet the requirement of the compositional language qualitatively.
> And it was never our intention to equate “compositionality” and “being able to do zero-shot test”, since we did define compositionality as “we express our intentions using words as building blocks to compose a complex meaning”. In the revised paper, we changed any sentence that might confuse the readers to think that we equate “compositionality” and “being able to do zero-shot test”, because we certainly do not.
>
> - Citation-less fifth sentence of section 4:
> Thank you for pointing this out. We added proper citations for this.
>
> - If this paper wishes to make any claims about compositionality, it must at a minimum:
> We would like to point out that ICLR is not a linguistics conference (nor is it a language-oriented one such as ACL) and our intention was not to develop a rigorous, quantifiable definition of compositionality and its evaluation strategy, for which even linguists have not been successful yet. The recent papers demonstrating the emergence of grounded/compositional communication using neural agents use a relatively relaxed definition of compositionality. But that does not make their work less interesting or meaningful. In the context of learning representations (which is the main theme of ICLR), we care about certain aspects more (e.g. pushing the limits of neural agents) and other aspects less (e.g. coming up with a definition of compositionality that is both linguistically and computationally meaningful). In this spirit, we believe our work has shown sufficient contributions in several aspects as described in the introduction (section 1).
> Lastly, the researchers in this field (neural agents and communication) have just begun to discuss the need and strategy for quantitative measure of how much the artificial language resembles human language (in aspects such as compositionality), and we believe our discussion in the last section raises relevant issues in accordance with the movement. Over time, such discussion will result in a practical, quantifiable measure of compositionality that most researchers can agree upon.
>
> - Claim for compositionality would be convincing if we can predict the string to be appeared in the gray boxes in Figure 3.
> We would like to point out that the messages in the gray boxes of Figure 3 do actually follow the patterns nicely, except for yellow ellipsoid. After seeing all blue objects and all box objects except blue box, we can infer that the blue box will be very likely be described by “eeeeee ee”. Granted, the compositional patterns in Table 3 are weaker than Table 2, but it is undeniable that there are patterns.

---

### Official Review · AnonReviewer3 · 2017-11-26
**The paper is well-written, clearly illustrating the goal of this work and the corresponding approach. The “obverter” technique is quite interesting since it incorporates the concept from the theory of mind which is similar to human alignment or AGI approach.**

**Rating:** 6
**Confidence:** 3

**Review:**

Pros:
1. Extend the input from disentangled feature to raw image pixels
2. Employ “obverter” technique, showing that it can be an alternative approach comparing to RL
3. The authors provided various experiments to showcase their approach

Cons:
1. Comparing to previous work (Mordatch & Abbeel, 2018), the task is relatively simple, only requiring the agent to perform binary prediction.
2. Sharing the RNN for speaking and consuming by picking the token that maximizes the probability might decrease the diversity.
3. This paper lack original technical contribution from themselves.

The paper is well-written, clearly illustrating the goal of this work and the corresponding approach. The “obverter” technique is quite interesting since it incorporates the concept from the theory of mind which is similar to human alignment or AGI approach. Authors provided complete experiments to prove their concept; however, the task is relatively easy compared to Mordatch & Abbeel (2018). Readers would be curious how this approach scales to a more complex problem.

While the author sharing the RNN for speaking and consuming by picking the token that maximizes the probability, the model loses its diversity since it discards the sampling process. The RL approach in previous works can efficiently explore sentence space by sampling from policy distribution. However, I can not see how the author tackles this issue. One of the possible reason is that the task is relatively easy, therefore, the agent does not need explicitly exploration to tackle this task. Otherwise, some simple technique like “beam search” or perform MC rollout at each time could further improve the performance.

In conclusion, this paper does not have a major flaw. The “obverter” approach is interesting; however, it is originally proposed in Batali (1998). Generating language based only on raw image pixels is not difficult. The only thing you need to do is replacing FC layers with CNN layers. Though this paper employs an interesting method, it lacks some technical contribution from themselves.


[1] Igor Mordatch and Pieter Abbeel. Emergence of grounded compositional language in multi-agent populations. AAAI 2018

[2] John Batali. Computational simulations of the emergence of grammar. Approaches to the evolution of language: Social and cognitive bases, 405:426, 1998.

---

> ### Author Response · Authors · 2018-01-04
> **Authors' response to the reviewers' comments.**
>
> We thank you for your thoughtful review. We address each of your concerns as follows.
>
> 1. The task is relatively simple:
> To the best of our knowledge, this is the first attempt to observe the emergence of compositional communication based on pure pixels. Our aim was to test the possibility of the emergence of compositionality based on a straightforward task so that we can perform extensive analysis on the outcome confidently, based on full control of the experiment. Especially, since our experiment maps a sequence of symbols to each factor (color and shape), instead of mapping one symbol to each factor (such as Mordatch & Abbeel 2017 [1] or Kottur 2017 [2]), rashly taking on complex tasks would have made the qualitative analysis of the emergent language exponentially difficult. As mentioned in our paper, a systematic method to evaluate the compositionality of a given language is an on-going effort in the emergent communication community, and as the evaluation strategy advances we will be more prepared to take on interesting, complex tasks to encourage the emergence of human-like language. We clarified our motivation for choosing a relatively simple task in the revised paper.
>
> 2. Using a single RNN for both speaking and listening hinders the agents’ chance to explore:
> While it is true that our deterministic approach in the paper might not match the typical exploratory behavior of RL, there are many ways to encourage the agents to explore the message space while using the obverter strategy. In fact, since only the listener’s parameters are updated and the speaker’s parameters are fixed during training, we can be quite creative with the message generation process and still be able to use gradient descent. One of the straightforward ways to increase the message diversity is to sample the character from a multinomial distribution at each timestep, instead of deterministically selecting the one that maximizes the output (y_hat). We can go a little further and adjust the temperature of the softmax function, so that the agents explore the message space a little more actively during the early training rounds and gradually convert to the deterministic behavior. As you suggested, this might help the agents discover a more optimal language when facing complex tasks. We reflected this comment in our revised paper.
>
> 3. This paper lacks original technical contribution:
> This may seem so based on our relatively light-weight model structure, and the fact that we employed the obverter strategy. The aim of the paper was to guide the agents to develop compositional communication, and we have shown strong evidence that it could be done with our approach, which combines the recent, advanced neural network techniques and a philosophy well motivated by psychology and linguistics. We believe proposing a method that solves an interesting problem (although, in our case, how successfully we solved the emergence of compositionality is hard to quantify) is itself a technical contribution, although the amount of contribution is subject to opinion. Also, the obverter strategy is not a specific algorithm like ADAM, but more like a general philosophy such as GAN, which has been used in various works besides Batali 1998 [3]. And we believe that introducing a successful philosophy from a different, yet relevant field to the machine learning community is a valuable contribution. We added a small description in the revised paper regarding how obverter strategy was originated in Hurford 1989 [4] and has been used in many works including Kirby and Hurford 2002 [5].
>
> [1] Igor Mordatch and Pieter Abbeel. Emergence of grounded compositional language in multi-agent populations. AAAI 2018
> [2] Satwik Kottur, Jose MF Moura, Stefan Lee, and Dhruv Batra. Natural language does not emerge ’naturally’ in multi-agent dialog. EMNLP 2017
> [3] John Batali. Computational simulations of the emergence of grammar. Approaches to the evolution of language: Social and cognitive bases, 405:426, 1998
> [4] James R Hurford. Biological evolution of the saussurean sign as a component of the language acquisition device. Lingua, 77(2):187–222, 1989
> [5] Simon Kirby and James R Hurford. The emergence of linguistic structure: An overview of the iterated learning model. In Simulating the evolution of language, pp. 121–147. Springer, 2002

---

### Decision · Program_Chairs · 2018-01-29
**ICLR 2018 Conference Acceptance Decision**

**Decision:**

Accept (Poster)

**Comment:**

This paper investigates emergence of language from raw pixels in a two-agent setting. The paper received divergent reviews, 3,6,9. Two ACs discussed this paper, due to a strong opinion from both positive and negative reviewers. The ACs agree that the score "9" is too high: the notion of compositionality is used in many places in the paper (and even in the title), but never explicitly defined. Furthermore, the zero-shot evaluation is somewhat disappointing. If the grammar extracted by the authors in sec. 3.2 did indeed indicate the compositional nature of the emergent communication, the authors should have shown that they could in fact build a message themselves, give it to the listener with an image and ask it to answer. On the other hand, "3" is also too low of a score. In this renaissance of emergent communication protocol with multi-agent deep learning systems, one missing piece has been an effort toward seriously analyzing the actual properties of the emergent communication protocol.  This is one of the few papers that have tackled this aspect more carefully. The ACs decided to accept the paper. However, the authors should take the reviews and comments seriously when revising the paper for the camera ready.